# Protocol for validation of the Global Scales for Early Development (GSED) for children under 3 years of age in seven countries

Vanessa Cavallera ,[1] Gillian Lancaster,[2] Melissa Gladstone ,[3] Maureen M Black ,[4,5] Gareth McCray ,[2] Ambreen Nizar,[6] Salahuddin Ahmed ,[7] Arup Dutta,[8] Romuald Kouadio E Anago,[9] Alexandra Brentani,[10] Fan Jiang ,[11] Yvonne Schönbeck,[12] Dana C McCoy,[13] Patricia Kariger,[14] Ann M Weber ,[15] Abbie Raikes,[16] Marcus Waldman,[16] Stef van Buuren ,[12,17] Raghbir Kaur,[1] Michelle Pérez Maillard,[1] Muhammad Imran Nisar ,[6] Rasheda Khanam ,[18] Sunil Sazawal,[8] Arsène Zongo,[9] Mariana Pacifico Mercadante,[10] Yunting Zhang,[19] Arunangshu D Roy ,[7] Katelyn Hepworth,[16] Günther Fink,[20] Marta Rubio-Codina ,[21] Fahmida Tofail,[22] Iris Eekhout ,[12] Jonathan Seiden,[13] Rebecca Norton,[1] Abdullah H Baqui,[18] Jamila Khalfan Ali,[23] Jin Zhao,[24] Andreas Holzinger,[25] Symone Detmar,[12] Samuel Nzale Kembou,[9] Farzana Begum ,[6] Said Mohammed Ali,[26] Fyezah Jehan ,[6,27] Tarun Dua,[1] Magdalena Janus [28]

For numbered affiliations see end of article.

**Correspondence to**
Dr Vanessa Cavallera;
cavallerav@who.int

## ABSTRACT

**Introduction** Children's early development is affected by caregiving experiences, with lifelong health and well-being implications. Governments and civil societies need population-based measures to monitor children's early development and ensure that children receive the care needed to thrive. To this end, the WHO developed the Global Scales for Early Development (GSED) to measure children's early development up to 3 years of age. The GSED includes three measures for population and programmatic level measurement: (1) short form (SF) (caregiver report), (2) long form (LF) (direct administration) and (3) psychosocial form (PF) (caregiver report). The primary aim of this protocol is to validate the GSED SF and LF. Secondary aims are to create preliminary reference scores for the GSED SF and LF, validate an adaptive testing algorithm and assess the feasibility and preliminary validity of the GSED PF.

**Methods and analysis** We will conduct the validation in seven countries (Bangladesh, Brazil, Côte d'Ivoire, Pakistan, The Netherlands, People's Republic of China, United Republic of Tanzania), varying in geography, language, culture and income through a 1-year prospective design, combining cross-sectional and longitudinal methods with 1248 children per site, stratified by age and sex. The GSED generates an innovative common metric (Developmental Score: D-score) using the Rasch model and a Development for Age Z-score (DAZ). We will evaluate six psychometric properties of the GSED SF and LF: concurrent validity, predictive validity at 6 months, convergent and discriminant validity, and

test–retest and inter-rater reliability. We will evaluate measurement invariance by comparing differential item functioning and differential test functioning across sites.

**Ethics and dissemination** This study has received ethical approval from the WHO (protocol GSED validation 004583

## STRENGTHS AND LIMITATIONS OF THIS STUDY

⇒ The study collects validation data (n=8736 children) for the Global Scales for Early Development (GSED) in seven countries that vary in geographic, linguistic, cultural and sociodemographic characteristics.

⇒ The methods for the validation of GSED are systematic across sites and follow rigorous standard operating procedures based on the best scientific evidence available.

⇒ A tablet-based App is used for data collection to make the administration of the GSED measures user-friendly, to reduce recording and transcribing errors and to facilitate adaptive testing.

⇒ The GSED short form and long form aim to include items that are culturally neutral and fit the Rasch model, which assume that child development milestones are age-ordinal, to create D-scores while psychosocial items are included in a separate measure (GSED psychosocial form (PF)) and cultural-specific items can be supplemented by countries.

⇒ The three secondary aims (preliminary reference scores, an adaptive testing algorithm and the feasibility and validity of the GSED PF) are exploratory and will require further research.

20.04.2020) and approval in each site. Study results will be disseminated through webinars and publications from WHO, international organisations, academic journals and conference proceedings.

**Registration details** Open Science Framework https://osf.io/ on 19 November 2021 (DOI 10.17605/OSF.IO/KX5T7; identifier: osf-registrations-kx5t7-v1).

## INTRODUCTION

Prenatal and early postnatal experiences have significant impacts on early childhood development (ECD) and can influence the accrual of health, well-being and productivity throughout the life course.[1] To promote current and sustainable peace and prosperity, the United Nations has focused the Sustainable Development Goals (SDGs) on improving children's outcomes in the early years through multiple targets. The most explicit target for young children is SDG 4 (Education goal), which requires reporting on the 'proportion of children under 5 years of age who are developmentally on track in health, learning and psychosocial well-being, by sex'.[2]

There are few valid measures that can be used globally to assess child development for children under 3 years of age. Current measures of ECD range from proxy measures (eg, prevalence of country-level stunting and poverty) to detailed measures of individual performance on developmental tasks.[3] The Early Childhood Development Index 2030 (ECDI 2030)[4] does not include children below 2 years of age. A recent review has identified the creation and validation of population-based instruments for assessing very young children as a global priority.[5]

The Global Scales for Early Development (GSED) build on advances made by analyses of existing global datasets,[6] and new data collection[7] that demonstrated the cross cultural applicability of items that measure young children's development. Three research teams[8] joined efforts to develop the GSED in response to the pressing need for instruments and metrics to measure ECD at population and programmatic levels across diverse parts of the world.

### The GSED

The GSED consist of three open-access measures developed by a WHO-led team to provide a standardized methodology for measuring the development of children aged 0-3 years (0-36 months) across diverse cultures and contexts.[9][10] They were developed for two objectives: 1) for population-level evaluation and 2) for programmatic evaluation through the caregiver report GSED Short Form (SF) and/or directly administered Long Form (LF). Additionally, a caregiver report GSED psychosocial form (PF) was created for measuring psychosocial behaviours. The development and piloting of the GSED SF, LF and PF are described elsewhere.[9–11]

The GSED SF and LF produce metrics on the same age-ordinal scale and quantify the same latent construct. The Developmental Score (D-score) (see box 1) underlies both measures and reflects children's overall development across multiple domains typically demonstrated in this age group (eg, cognitive, motor, language, social-emotional).[6]

> **Box 1    The Developmental Score (D-score)**
>
> The Developmental Score,[15] or D-score, is a unidimensional latent variable measuring child development during the first 3 years across multiple domains. The milestones that make up the D-score conform to the Rasch model,[28] thus yielding a scale with interval properties with a fixed unit (figure 1). It is therefore possible to calculate a meaningful difference between two D-scores. Similar to height-for-age Z-score, given suitable age-conditional references, the D-score can be transformed to a Z-score that accounts for children's age (ie, Development for Age Z-score, or DAZ). The DAZ facilitates comparisons across children of different ages.

The GSED PF items, designed to measure non-normative developmental patterns, including behavioural or regulatory challenges, are not age-ordinal and do not use the D-score metric.

### AIMS

The primary aim of this study is to validate the GSED measures,[11] through testing for measurement invariance and evaluation of the psychometric properties to measure development among children under 3 years (<36 months) globally (including creation of D-scores and Development for Age Z-score (DAZ)).

Specific objectives:
1. Fit a Rasch model to the item data to calculate the D-scores and DAZ.
2. Investigate differential item functioning (DIF) and differential test functioning (DTF) across sites to determine measurement invariance.
3. Examine psychometric properties of the GSED SF and LF:
   – Test–retest and inter-rater reliability (score and item level).
   – Concurrent validity (association between scores on GSED and Bayley Scales of Infant and Toddler Development (Bayley-III) or Griffiths Scales of Child Development administered concurrently).[12]
   – Convergent validity (strength of association between GSED D-scores and other theoretically relevant constructs).
   – Predictive validity (association between GSED scores 6 months after initial assessment).

The secondary aims are to: (1) establish preliminary reference scores (The population has not been selected as a representative sample of all children aged <3 years in each site (as would happen in a countrywide population census). Selection and recruitment of a representative sample was beyond the scope of this study and not required for validation purposes. We are therefore developing 'reference scores' which should not be interpreted as population-sampled norms.) for optimal development on the D-score (GSED SF and LF), (2) develop and validate an adaptive testing algorithm and (3) obtain preliminary validity data on the psychometric properties of the GSED PF.

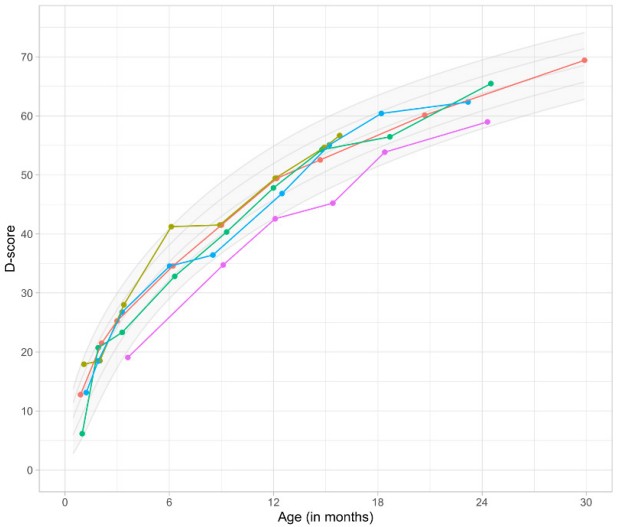

**Figure 1** Development chart. Reproduced with permission from Van Buuren and Eekhout.[15]

## METHODS
### Design and study sites
The GSED validation study uses a prospective cross-sectional design with a longitudinal component of age and sex stratified samples of children in seven countries. The countries are culturally, linguistically and geographically diverse, representing low-income (Bangladesh, Côte d'Ivoire, Pakistan, United Republic of Tanzania), middle-income (Brazil and The Republic of China) and high-income (The Netherlands) settings. Samples in each site are not nationally representative; however, they are diverse, for example, covering both rural/urban settings.

Preparation and feasibility phases are described elsewhere,[11] and assess feasibility of administration of GSED and associated measures including processes for translating and culturally adapting GSED and other study measures, creating data management systems and training teams in data collection procedures.

### Patient and public involvement
Caregivers of children 0–41 month-olds were involved in the study design as the burden of the assessment was discussed with them in a pilot stage through qualitative data collection. We intend to disseminate the main results to trial participants and will seek patient and public involvement in the development of an appropriate method of dissemination.

### Study sample
The study sample includes children between 0 and 41 months of age (inclusive) living in study areas (see table 1 for inclusion and exclusion criteria). The small sample of children from 36 to 41 months aims to ensure that parameters are estimated with adequate precision for children at the top of our age range (36 months).

### Recruitment and consent
In each site, the sampling frame consists of a list of potentially eligible caregiver–child dyads residing in the defined study area. Lists of potential participants are created in compliance with ethical review boards approved processes; they vary by site and may include: participants in local pregnancy surveillance systems, families who have previously agreed to be contacted for participation, from hospital/health centre registries or families with children attending local child health/care centres. Sites using registries will rely on hospital or health centre staff (unaffiliated with GSED) to contact families and obtain consent for sharing their information with the GSED team. A sample listing of the pre-consented families will be provided to the GSDE team for recruitment. Sites recruiting families from local child health/care centres will rely on advertisements or flyers with information about the project, participation requirements, GSED team contact information for questions, and a scan code or website link for interested families to provide basic eligibility information and consent to be contacted for enrolment.

Eligible children are sampled from this list using the GSED sampling scheme (figure 2). To minimise clustering of correlated scores within households, one child per caregiver and in multi-family household is selected, guided by age and sex quotas. For siblings or twins, one is chosen randomly. Target children's primary caregiver (person most familiar with the child and spends most time with them) is approached for consent and enrolment. A non-technically worded information sheet is shared and consent to participate is obtained at first visit. In the Netherlands, participants provide consent online, confirmed by study staff at first visit. Refusals to participate and dropouts are registered and replaced.

| Table 1 | Study sample inclusion and exclusion criteria | |
| --- | --- | --- |
| **Sample** | **Inclusion criteria** | **Exclusion criteria** |
| Total per site n=1248 (as described in sample size section below) | 1. Age 0–41 months<br>2. Family speaks to the child in same language as GSED translation<br>3. Primary caregiver available to participate | 1. Missing gestational age (ultrasound or last menstrual period (LMP))<br>2. Missing birth weight data<br>3. Acutely unwell at time of assessment (temporary exclusion: to be rescheduled after 7 days) |
| GSED, Global Scales for Early Development. | | |

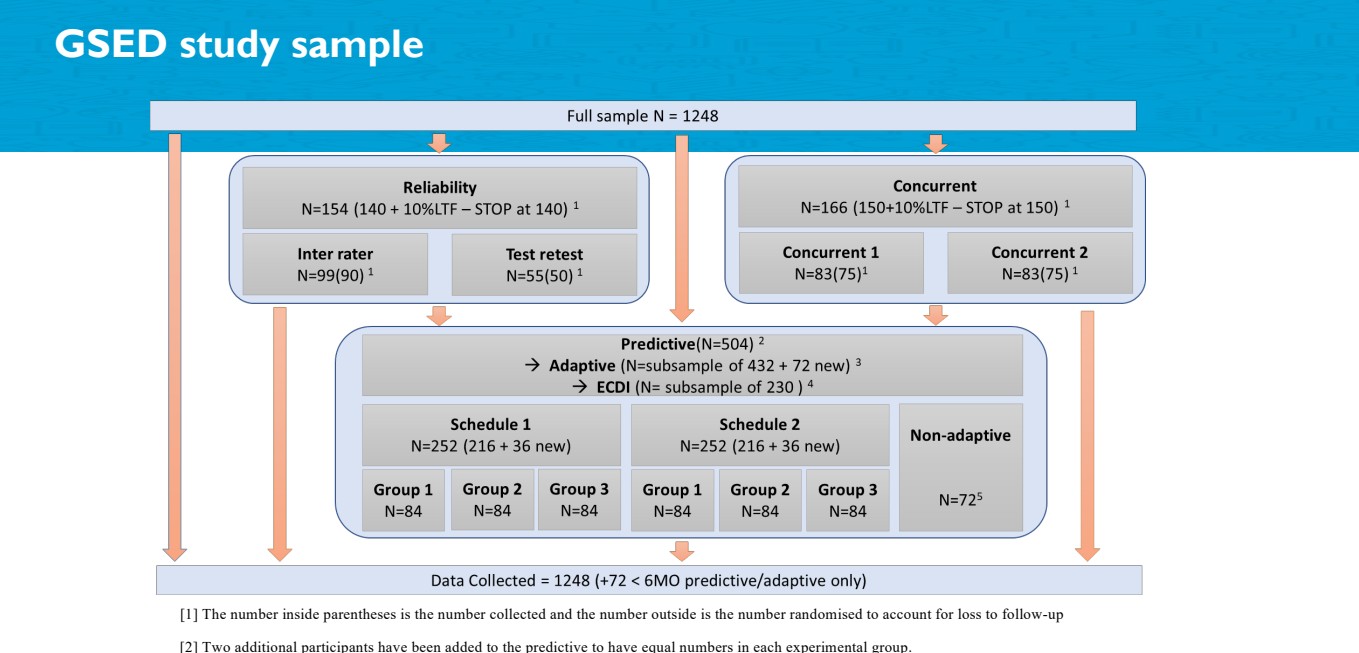

**Figure 2** Study sampling schema diagram. ECDI, Early Childhood Development Index; GSED, Global Scales for Early Development.

## Sampling frame and schemes

Sample size for recruitment within each site is 1248 children (total 8736 children) across seven countries. After consent is provided, children are allocated by sex and age groups using a randomization procedure to one of several sampling schema (eg, predictive, reference-score, reliability; figure 2). See sampling in online supplemental table S1 for sampling frame. Out of the full site sample of 1248 children, 504 children per site are randomly selected for re-evaluation 6 months later to assess predictive validity (primary aim). A second scheme indicates the minimum subsample of children needed to calculate preliminary reference scores (secondary aim) that will facilitate cross-country comparisons. To maximise precision of parameter estimates, larger quotas are kept for the youngest age brackets where rates of development are accelerated. A third scheme addresses inter-rater reliability for 90 children per site using two assessors who independently assess the same child sequentially or within 24 hours[12] (We note that this procedure differs from typical inter-rater reliability (IRR) designs which involve simultaneous scoring of a single assessment. This sequential design was necessitated by logistical constraints. Given that this design captures both variance due to differences in raters and differences in occasions, the observed IRR represents a lower bound for the true inter-rater reliability of the assessments.). Test–retest (intra-rater reliability) is performed by inviting 50 children per site to return for repeat assessment with the same rater within 7–10 days. For concurrent validity, to assess the GSED against the Bayley-III, a sample size of n=150 per country produces a two-sided 95% CI 0.15 to 0.44, when the estimate of Pearson's product-moment correlation is 0.3, with an equal spread of participants tested across age and sex.

In the Netherlands, the GSED SF and PF are administered online. A subset of participants (n=32) are interviewed face-to-face to compare method of administration. To determine test–retest reliability (intra-rater reliability), the primary caregiver completes the SF and PS form online and then a second time 7–10 days later.

## Data collection

### Measures

#### GSED

#### GSED SF and LF

The creation of the GSED SF and LF is described elsewhere.[10] Briefly, we constructed an item bank from previously gathered data and compiled cross-sectional and longitudinal data from 31 countries representing over 73 000 anonymised children with 109 079 assessments (using 22 established ECD instruments).[6 13 14] Using subject matter expert input and statistical modelling,[15] we developed two measures intended to capture child development at population-level and/or to evaluate programmatic impacts: a caregiver-reported measure (GSED SF), and a directly administered measure (GSED LF).[10] The measures are created paper-based and app-based (GSED App) with built-in administration rules and supporting media-files (see below).

The GSED SF includes 139 items representing emerging skills and behaviours within cognitive, motor, language and social-emotional domains. All items are presented as questions to the caregiver, with binary response options (Yes/No and 'Don't Know') that use start rules based on the child's age, and stop rules based on age and performance. Assessors record caregiver's responses, regardless of the assessor's observations. In the Netherlands only, the GSED SF is completed online by caregivers. The GSED SF administration includes sounds, images and short video clips that assist in understanding, interpretating and administering the items.

The GSED LF includes 155 items capturing similar domains to the SF but, observed by the assessor following start and stop rules based on the child's age and responses. LF items must either be observed incidentally or by eliciting the behaviour or both, depending on the item. Items are organised into three grids (A, B and C) that enable assessors to measure the child's performance on similar tasks in succession, making the administration easier for both assessors and children. To further facilitate administration, icons are placed next to each item that inform the assessor whether the item is observed, demonstrated to or by the child, listened for or spoken to the child. The GSED LF uses a locally constructed and low cost kit with basic materials that the child interacts with to demonstrate abilities. The kit is created by local teams with detailed guidance from WHO. Responses of all LF items are binary (skill observed/not observed).

The items in both measures are ordered by difficulty reflecting children's emerging skills. Based on the analyses from the validation, we will select the items to be included in the final GSED SF and LF versions available for use.

### Psychosocial form (PF)

Unlike the SF and the LF, the GSED PF has been developed to index non-normative developmental patterns that provide a window into early manifestations of children's mental health challenges, including internalising and externalising behaviour problems and dysregulation (eg, eating and sleeping). Items capturing developmentally normative information about socio-emotional competencies are included in the GSED SF and LF, as the SDG 4.2 includes children's psychosocial wellbeing. Because few instruments have been developed to capture psychosocial difficulties for children under 3 years, little existing data are available and the development of the GSED PF is exploratory. The PF initial prototype was created through a review of existing measures of infant and toddler mental health and consensus by subject matter experts. The GSED PF includes 47 items and reflects caregiver perceptions of the behaviours' frequency, using response options: often; sometimes; never/almost never. Items are divided into two age groups: 0 to <6 and 6 to <36 months.

### Contextual and demographic measures

In addition to the GSED, the validation study includes measures of children's growth and nutrition, health, environmental and contextual information (see table 2 for measures and sources). The selection of measures was based on known biological and social determinants of development,[16] the demonstrated validity of the contextual measures in at least one low/middle-income country, and efficiency for data collection. See online supplemental file S2 for visit schedules (online supplemental tables S2A and S2B).

In three sites (Côte d'Ivoire, The Netherlands and The People's Republic of China) where administration of the Home Observation for Measurement of the Environment Inventory is not feasible, household stimulation data and caregiver–child activities are collected using Family Care Indicators. In all sites, a concurrent measure of child development (Bayley-III or Griffiths Mental Development Scales) is administered in a subsample of children to determine concurrent validity of GSED to a well-established measure of the same construct.

### Schedule

Data collection is scheduled over one to three visits depending on the study site to accommodate rules of measure administration order and location. The first administration of the GSED SF and PF is completed at home (or online in the Netherlands) to test it in the setting intended for future use (eg, Multiple Indicator Cluster Surveys or Demographic and Health Surveys) and prior to administration of the GSED LF. The GSED LF is administered in a controlled environment (eg, clinic) to match the required concurrent validity testing protocols. For the concurrent validation, the GSED and concurrent measures are administered in the same location on different days and counter-balanced in order of administration.

### Training and quality control

Training of local master trainers is performed by the WHO team for the GSED SF, PF and LF, using slide presentations, discussion forums, audio–visual aids and practice exercises. Local master trainers are responsible for training local field teams using materials adapted and translated to local languages. Reliable administration of the GSED measures must be met (inter-rater agreement with a master trainer of ≥90%) for certification.

To ensure quality assurance, 10% of all the study visits are observed by the study supervisor in person (or through video-recording in the Netherlands), covering each child age band and certified assessors. Supervisors independently complete questionnaires being administered by the assessor and complete a fidelity checklist. Assessors are given feedback based on checklist score. Supervisors review quality assurance findings with the WHO biweekly, along with discussions with the subject matter experts for further resolution, as needed.

**Table 2**  Study measures in addition to GSED

| Construct | What the measure captures | Measure | Administration mode | Time for administer (min) |
|---|---|---|---|---|
| Child health and household socioeconomic status (SES) | ▶ Eligibility (exclusion criteria)<br>▶ Demographic information<br>▶ Information about acute child health<br>▶ Delivery and perinatal conditions<br>▶ Breast feeding<br>▶ Child's health history<br>▶ Household socioeconomic status*<br>▶ Caregiver education<br>▶ Maternal health/chronic illness<br>▶ COVID-19 exposure | Eligibility and contextual form (specifically developed for the study) | Caregiver report | 35 |
| Anthropometry | ▶ Weight at time of assessment<br>▶ Infant length/child height at the time of assessment<br>▶ Child's mid-upper arm circumference at time of assessment<br>▶ Child's head circumference at time of assessment | Anthropometry form | Child assessment | 15 |
| *Family/home environment* | ▶ Home environment (HOME only)<br>▶ Play/stimulation/interactions between the child and other family members in the home (HOME and FCI) | Home observation for measurement of the environment inventory (HOME)[29] OR family care indicators (FCI)[30] † | *HOME*: caregiver report & observation *FCI*: caregiver report | *HOME*: 45 *FCI*: 15 |
|  | ▶ Child neglect/abuse<br>▶ Exposure to violence or conflict | Childhood Psychosocial Adversity Scale (CPAS)[31] † | Caregiver report | 15 |
|  | ▶ Family resilience | Brief Resilience Scale (BRS)[32] † | Caregiver report | 1 |
|  | ▶ Family social support | Family Support Scale (FSS)[33] † | Caregiver report | 5 |
| Caregiver health and well-being | ▶ Caregiver depressive symptoms | The Patient Health Questionnaire- 9 (PHQ-9)[34] | Caregiver report | 5 |
| Child development | ▶ Global child development (0–41 months) | Bayley Scales of Infant and Toddler Development (Bayley-III)[35] OR Griffiths Mental Development Scales[36] ‡ | Direct child assessment | 45–60 |
|  | ▶ Global child development (24–41 months) | Early Childhood Development Index 2030 (ECDI2030)[4] § | Caregiver report | 10 |

*Socioeconomic information on this form comes from the standard DHS multiple assets index; however, some sites have adapted the socioeconomic status items to better fit their contexts.
†These measures have been minorly adapted for the purpose of the study.
‡In a subsample (n=150).
§In a subsample (all children of 24–41 months within the predictive validity subsamples in three countries).
DHS, Demographic and Health Survey; GSED, Global Scales for Early Development.

The GSED application software for data collection has built-in data range and consistency checks. Data managers review and resolve issues daily in consultation with the local field team and/or WHO team.

**Sample size**

Sample size determination was based on the primary aim of assessing the psychometric properties of the GSED. To have sufficient power to estimate measurement

parameters (abilities and difficulties) needed to calculate the D-score and DAZ scores at baseline and to detect DIF of 1 logit with a power of $1-\beta=0.90$ and a two-sided significance level of $\alpha=0.05$, a sample of n=1248 per site is required. Given the rapidity of development of children at this age, the latent trait is longer than tends to be found in educational tests which focus on a narrower ability range. The easiest item in our tool 'Does your child smile?' has a difficulty of −13.2 logits (1.1 on the D-score scale) and the most difficult item has a difficulty of 8.4 logits (88.86 on the D-Score scale), a 21.6 logit span. Thus, a one logit difference is not particularly large, given the length of the latent trait. This sample size was calculated via optimisation of the sample size at (a) each age/sex stratum and (b) overall on 1000 simulated datasets generated from parameters suggested by the Rasch GSED model. See online supplemental file S1 for additional details.

### Statistical analysis

To construct the scores for the GSED SF and LF, a Rasch model will be fitted and the item fit statistics (infit and outfit) will be assessed.[17] Any items with unacceptable fit levels will be removed. Items will be screened for whether they exhibit unacceptable levels of measurement non-invariance (ie, they have approximately equal difficulties) across countries and other contextual variables. Items exhibiting unacceptable DIF (using the logistic regression method) will be discarded sequentially, and the item response models will be refit using the remaining items. The expected a posteriori (EAP) method[18] will be applied to the final model to estimate the latent ability parameter (the D-score). Systematic deviations from unidimensionality will be tested by performing a principal components analysis on the residuals of the Rasch model. The method uses a prior normal distribution with a mean set equal to the average proficiency at the child's age and an SD of 5. The ability estimates will be used to estimate preliminary developmental percentile curves against age using a Generalized Additive Model for Location Scale and Shape (GAMLSS). Note that this application of EAP estimates underestimates the true variability in the population because EAP estimates—as any measurement—are always imprecise. In daily practice, analysts will compare other EAP estimates to the reference. To support this type of application, we create the references from the EAP estimates and accept a (perhaps slight) underestimate of the true variability in child development in the population. Following previous methodology,[19] software will be written to calculate DAZ-scores based on the final dataset in R,[20] and a user-friendly front-end version created in R (ShinyApp)[21] and/or Excel.

Reliability (inter-rater and test–retest) for all GSED measures will be analysed using intraclass correlation coefficient (at the score level) and Gwet's AC1 agreement (at the item level) statistics with 95% CIs to determine whether items perform reliably within and between assessors.[22] A cut-off value of 0.4 and above will be used to flag items as adequately reliable. Those items with agreement between 0.4 and 0.5 will be discussed to determine if modifications can be made to improve their administration and/or comprehension.

DAZ scores from the GSED SF and LF will be used to conduct validity analyses to ensure that the measures are capturing the construct they are purported to measure (construct validity). Concurrent validity will be assessed by correlating age-corrected Bayley-III or Griffiths Mental Development Scales scores with GSED DAZ scores. We anticipate that these scores will have low to moderate positive correlations. Convergent validity will be supported by statistically significant positive correlations (with 95% CI) between the GSED scores and continuous contextual measures with prior evidence of association with child development. Comparisons between 'known groups' will be made using the following variables: maternal education, home learning opportunities, home environment, socioeconomic status, maternal mental health and child anthropometry, and stunting to determine if scores discriminate between high and low categories for each variable using mean DAZ scores.

GSED scores at baseline and follow-up will be correlated for predictive validity (positive association between baseline and at 6 months) and mixed-effects linear regression used to adjust for other contextual covariates and baseline scores.

### Secondary (exploratory) aims
#### Reference scores

We plan to develop a set of preliminary reference scores to facilitate comparison of DAZ scores across countries. From the full validation study sample, a subsample of children who have not experienced prior exposure to major known biological and environmental risk factors is selected (ie, 'reference subsample') (table 3). Such an approach relies on the assumption that the attainment of basic developmental milestones captured by the GSED of children who are free of major risk factors is relatively similar globally.[23]

To develop the reference scores, we will fit GAMLSS[24] to flexibly model both conditional means, conditional SD of scores, and, if necessary, conditional skewness and kurtosis. We will test our assumption that the distribution of scores is equivalent across sites by adding a site indicator at each moment of the distribution, and testing site effects for their statistical significance. Where possible, we will conduct standardisation of scores to assist with the interpretation of scores by pooling data across countries. We will report the corresponding parameters of the GAMLSS model at appropriate ages.

#### Adaptive testing

We will determine whether adaptive testing is a feasible and valid option to measure child development within the GSED (box 2). Adaptive testing[25] is an administration method that continually adapts to the level of the child's performance, thereby reducing test administration

**Table 3** 'Reference' subsample exclusion criteria (healthy subsample)

| Sample | Exclusion criteria |
|---|---|
| Minimum subsample of 'reference' children per site n=522 | 1. Below secondary maternal education (<6 years of schooling)<br>2. Birth weight <2500 g<br>3. Gestational age <37 completed weeks (259 days) and ≥42 completed weeks (294 days) (assessed by ultrasound)<br>4. Undernutrition (weight for age, length for age, OR weight for height Z score of <–2 on the WHO Child Growth Standards) at the time of developmental assessment<br>5. Known severe congenital birth defect<br>6. History of birth asphyxia OR neonatal sepsis requiring hospitalisation<br>7. Known neurodevelopmental disorder/disability (Severe visual problems, seizures, hearing impairment) OR other chronic health problems (ie, congenital heart disease) |

time. Previous simulations[26] indicated that theoretically substantial gains in the precision of scores are possible when using adaptive testing even if administering fewer items.

### Psychosocial form

The PF measure is in an early stage and will undergo exploratory and confirmatory factor analyses to assess the internal scale structure. Associations between items and factor scores with variables suggesting a high risk of psychosocial stress, such as family resilience, social support, and family and community violence, in addition to GSED SF and LF scores (concurrent validity measures) will be examined.

### ETHICS AND DISSEMINATION

The study complies with the International Ethical Guidelines for Biomedical Research Involving Human Subjects[27] and received ethical approval from the appropriate body in each site (Bangladesh—Projahnmo Research Foundation Institutional Review Board; Brazil—University Hospital, São Paulo (HU-USP); Cote d'Ivoire—Comite National D'Ethique des Sciences de la Vie et de la

Sante (CNESVS); Pakistan—The Aga Khan University Ethics Review Committee; The Netherlands—Institutional Review Board TNO, Netherlands Organisation for Applied Scientific Research; The People's Republic of China—IRB of Shanghai Children's Medical Center Affiliated to Shangai Jiao Tong University School of Medicine; United Republic of Tanzania—Zanzibar Health Research Institute) and within WHO (protocol GSED validation 004583 approved on 20 April 2020). The findings of the study will be disseminated following a comprehensive dissemination strategy to reach a diverse range of stakeholders at the local, national and international level.

### DISCUSSION

The validation of the GSED SF and LF is a meticulous and systematic global process that introduces an innovative common metric (the D-score) that countries can use to track the progress of child development among populations of young children and will enable countries to adapt, modify and evaluate their policies and programmes to ensure that young children are effectively and equitably reaching their development potential and building the human capital needed for sustainable development. Additional attention is required on understanding young children's responses to psychosocial challenges within global contexts. The exploration of the GSED PF introduces an important opportunity to capture the non-normative developmental patterns among young children that are potential precursors to behaviour and psychiatric problems. The GSED validation has several important design, methodological and implementation characteristics that illustrate the rigour required to validate instruments to measure child development globally. First, it is conducted in seven countries with multiple linguistic, cultural and socioeconomic backgrounds. Second, GSED is implemented through an app-based data collection system that facilitates the implementation by reducing recording and transcribing errors and other common pitfalls of paper-based instruments. Third, this study builds on the best practices in validation by including a broad spectrum of psychometric methodologies (concurrent, predictive, convergent and discriminant validity, test–retest and inter-rater reliability, DIF and DTF). Fourth, a secondary aim builds the evidence for the creation of preliminary

---

**Box 2  Adaptive testing validation methodology**

We investigate the feasibility by applying adaptive testing in addition to the traditional 'fixed' GSED administration methods in the subsample designated for predictive validity analyses (n=502 per site) in three sites. The adaptive test is executed using tablets that are specially programmed to continually adjust child's score after each item is administered, and to suggest the next item based on the answers already received (eg, a more difficult item for a child with a higher score, an easier item for a child with a lower score). Once the programme establishes a reliable score, the administration is terminated. Both the adaptive test and the fixed test are administered with the same subsample during two separate visits alternating the order of administration to investigate the difference between the two modes of administration. We will investigate the following: the variance of user experience as a function of the average difficulty of milestones (leniency); the comparison of the D-score distribution under the adaptive testing procedure with the D-score distribution under the fixed GSED administration (using a z-test to assess the equivalence of the two modalities and plotting the results to show the level of concordance) and relation of the difference between the two D-scores to background variables.

reference scores for the SF and LF, based on a subsample with minimal exposure to major biological risk factors and to the extent possible, minimal social and environmental risk factors. Fifth, we are validating an adaptive testing design that can streamline administration by tailoring and reducing the number of items required to obtain a valid score. Sixth, we are testing a new measure of young children's non-normative psychosocial development.

One notable difference between the GSED SF and LF measures and other instruments of early child development is that the GSED measures are based on a unidimensional model of development through measurement approaches that are universally applicable across cultures. The measures do not follow the common multidimensional approach with separate scores for different domains or contexts. Our validation study intends to demonstrate that this model provides valid, reliable and interpretable data globally. The GSED SF and LF may exclude some items that measure development in cultural or setting-specific ways, because the focus is on selecting items that are meaningful for understanding child development within any given setting. If specific aspects need to be captured locally, to increase cultural relevance we suggest that the GSED measures are lightly adapted with country or culture-specific item props (in agreement with WHO) and/or through the administration of additional measures.

There are several limitations to our study. Although we are validating the GSED in seven countries, including one high income setting, three sites are resource-limited (Bangladesh, Pakistan and United Republic of Tanzania). Additional evidence may be needed in high income countries to expand the validity and reliability of the GSED to population-representative samples in additional countries. Second, the GSED has been created using items that fit a Rasch model demonstrating developmental progress across ages 0–3 years.[9] This univariate model makes strict assumptions and may exclude items that do not show strong age gradients or items that measure development in a culturally-specific ways. Third, GSED was developed to address population and programmatic level evaluations of early child development globally. The GSED is presently not being validated for screening or diagnosing individual children. Finally, our three secondary aims are exploratory, and will require further research, including developing global standards to replace our preliminary reference scores with more specific global norms, as in the Multi-country Growth Reference Standards for children's weight and height. In the future we plan to collect additional data from countries using strict inclusion/exclusion criteria (eg, additional considerations around environmental risk and protective factors) to further validate our initial reference scores. Similarly, we plan to conduct further work to explore the functionality, reliability, validity and invariance of the PF. Lastly, as the GSED SF and LF scores are meant to be interpreted and used for population-level measurement, we plan to expand the work towards understanding of how the GSED package

could be modified and validated to be able to identify individual children at risk of developmental delays and disorders.

**Author affiliations**
[1]Department of Mental Health and Substance Use, World Health Organization, Geneva, Switzerland
[2]School of Medicine, Keele University, Keele, UK
[3]Department of Women and Children's Health, Institute of Life COurse and Medical Sciences, University of Liverpool, Liverpool, UK
[4]International Education, RTI International, Research Triangle Park, North Carolina, USA
[5]Department of Pediatrics, University of Maryland School of Medicine, Baltimore, Maryland, USA
[6]Department of Paediatrics and Child Health, The Aga Khan University, Karachi, Sindh, Pakistan
[7]Research, Projahnmo Research Foundation, Dhaka, Bangladesh
[8]Center for Public Health Kinetics, CPHK Global, Pemba, Zanzibar, Tanzania
[9]IPA Côte d'Ivoire, Innovations for Poverty Action, Abidjan, Côte d'Ivoire
[10]Department of Pediatrics, University of São Paulo Medical School, São Paulo, Brazil
[11]Shanghai Children's Medical Center Affiliated to Shanghai Jiao Tong University School of Medicine, Shangai, People's Republic of China
[12]Department of Child Health, Netherlands Organization for Applied Scientific Research, Leiden, Netherlands
[13]Education Policy and Program Evaluation, Harvard Graduate School of Education, Cambridge, Massachusetts, USA
[14]Center for Effective Global Action, University of California Berkeley School of Public Health, Berkeley, California, USA
[15]School of Public Health, University of Nevada Reno, Reno, Nevada, USA
[16]Health Promotion, University of Nebraska Medical Center College of Public Health, Omaha, Nebraska, USA
[17]Department of Methodology and Statistics, Faculty of Social and Behavioural Sciences, University of Utrecht, Utrecht, Netherlands
[18]International Center for Maternal and Newborn Health, Department of International Health, Johns Hopkins Bloomberg School of Public Health, Johns Hopkins University, Baltimore, Maryland, USA
[19]Child Health Advocacy Institute, National Children's Medical Center, Shanghai Children's Medical Center Affiliated to Shanghai Jiao Tong University School of Medicine, Shanghai, People's Republic of China
[20]Swiss Tropical and Public Health Institute, University of Basel, Basel, Switzerland
[21]Social Protection and Health Division, Inter-American Development Bank, Washington, DC, USA
[22]Nutrition and Clinical Services Division (NCSD), International Centre for Diarrhoeal Disease Research Bangladesh, Dhaka, Bangladesh
[23]Research Division, Public Health Laboratory, Pemba, Zanzibar, Tanzania
[24]Department of Developmental and Behavioural Pediatrics, National Children's Medical Center, Shanghai Children's Medical Center Affiliated to Shanghai Jiao Tong University School of Medicine, Shangai, People's Republic of China
[25]IPA Francophone West Africa, Innovations for Poverty Action, Abidjan, Côte d\'Ivoire
[26]Institution Head, Public Health Laboratory, Pemba, Zanzibar, Tanzania
[27]Paediatrics and Child Health, The Aga Khan University, Karachi, Sindh, Pakistan
[28]Offord Centre for Child Studies, Department of Psychiatry and Behavioural Neurosciences, McMaster University, Hamilton, Ontario, Canada

**Contributors** All authors contributed substantively to this work. VC was the lead author in drafting the manuscript in addition to the technical contributions to the study protocol conceptualization and development. TD led the conceptualisation of the study, MG, MMB, MJ, and PK contributed significantly to the conceptualisation of the study design and methodology, drafted sections of the protocol and related manuscript; GL, GM (focus on psychometric properties), DCM, JS (focus on preliminary reference scores), MW (focus on testing of psychosocial form), SvB and IE (focus on adaptive testing) came to consensus on statistical analysis plan, determined the sample size calculations and drafted the related parts of the manuscript relevant to their specific expertise; AN, AR, KH and AW drafted substantial pieces of the manuscript related to sampling frame, study measures

and implementation. SA, AD, RKEA, ABr, JKA, FJ, YS, IN, RKa, SS, AZ, MPM, YZ, FT, ADR, ABa, JZ, AH, GF, SD, NSK, FB, SMA, FJ and MR-C contributed to the adaptation of the study protocol for feasibility and on-the-ground implementation, focusing on manuscript write up related to site-specific descriptions. All above authors, in addition to RK, MPM and RN reviewed and edited the study protocol and the manuscript. All authors read and approved the final manuscript submission.

**Funding** This work was supported (alphabetical order) by Bernard van Leer Foundation, Bill & Melinda Gates Foundation, Children's Investment Fund Foundation, Jacobs Foundation and King Baudouin Foundation, USA. The funders provided financial support. The design, implementation and writing of the manuscript were led by the WHO.

**Disclaimer** The author is a member of the WHO. The author alone is responsible for the views expressed in this publication and they do not necessarily represent the decisions, policy or views of the WHO (Applies to Cavallera V, Dua T, Kaur R, Pérez Maillard M and Norton R). The views here presented do not represent the Inter-American Development Bank, its board of directors, or the countries it represents (Applies to Rubio Codina M).

**Competing interests** None declared.

**Patient and public involvement** Patients and/or the public were involved in the design, or conduct, or reporting, or dissemination plans of this research. Refer to the Methods section for further details.

**Patient consent for publication** Not applicable.

**Provenance and peer review** Not commissioned; externally peer reviewed.

**ORCID iDs**
Vanessa Cavallera http://orcid.org/0000-0001-5623-9721
Melissa Gladstone http://orcid.org/0000-0002-2579-9301
Maureen M Black http://orcid.org/0000-0002-6427-4639
Gareth McCray http://orcid.org/0000-0002-0728-5171
Salahuddin Ahmed http://orcid.org/0000-0001-6771-0638
Fan Jiang http://orcid.org/0000-0003-0634-101X
Ann M Weber http://orcid.org/0000-0001-8130-5858
Stef van Buuren http://orcid.org/0000-0003-1098-2119
Muhammad Imran Nisar http://orcid.org/0000-0002-2378-4720
Rasheda Khanam http://orcid.org/0000-0002-9365-8594
Arunangshu D Roy http://orcid.org/0000-0002-6781-6325
Marta Rubio-Codina http://orcid.org/0000-0002-1286-7918
Iris Eekhout http://orcid.org/0000-0002-0030-1458
Farzana Begum http://orcid.org/0000-0002-8963-0230
Fyezah Jehan http://orcid.org/0000-0002-5874-4358
Magdalena Janus http://orcid.org/0000-0002-9500-6776

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
