## [Reviewer comments · BMJ Open]

ARTICLE DETAILS

TITLE (PROVISIONAL)	Protocol for Validation of the Global Scales for Early Development (GSED) for Children under 3 Years of Age in Seven Countries
AUTHORS	Cavallera, Vanessa; Lancaster, Gillian; Gladstone, Melissa; Black, Maureen; McCray, Gareth; Nizar, Ambreen; Ahmed, Salahuddin; Dutta, Arup; Anago, Romuald; Brentani, Alexandra; Jiang, Fan; Schönbeck, Yvonne; McCoy, Dana; Kariger, Patricia; Weber, Ann; Raikes, Abbie; Waldman, Marcus; van Buuren, Stef; Kaur, Raghbir; Maillard, Pérez; Nisar, Muhammad; Khanam, Rasheda; Sazawal, Sunil; Zongo, Arsène; Pacifico Mercadante, Mariana; Zhang, Yunting; Roy, Arunangshu; Hepworth, Katelyn; Fink, Günther; Rubio-Codina, Marta; Tofail, Fahmida; Eekhout, Iris; Seiden, Jonathan; Norton, Rebecca; Baqui, Abdullah; Zhao, Jin; Holzinger, Andreas; Detmar, Symone; Kembou, Samuel; Begum, Farzana; Jehan, Fyezah; Dua, Tarun; Janus, Magdalena

VERSION 1 – REVIEW

REVIEWER	Anita D'Aprano The University of Melbourne
REVIEW RETURNED	15-May-2022

GENERAL COMMENTS	Overall comment This manuscript describes a study protocol validating the global scales for early development (GSED) in seven countries. The creation of a measure that monitors children's early development is a priority and this is much needed research. The study proposes to recruit 1248 children per country and fit an item response model (specifically Rasch model) to the data to yield an interval measure (D-Score). The paper is very well written and clear. Introduction Clearly establishes the need for this measure and research. Aims Clear aim and detailed objectives that explain how the aim will be achieved. Methods Patient and Public involvement: Co-designed with participants. However, the participants are not described. Is this the caregiver or the services using the measure? This needs elaboration. Study sample: Inclusion criteria does not provide a justification for age up to 41 months. How was this age determined? Recruitment and consent: It is not clear if families could be approached from being identified through birth registries. Are their details shared with research team? Is this a breach of privacy? How
---

are families with children attending local child health/care centres approached? In a general 'advertisement' inviting participation? This needs further detail to ensure the process is ethical.

Sampling frame and schemes: indicate here that the supplementary file has this information.

Data Collection

The GSED LF is described as capturing similar domains to the SF, but observed by the assessor. If the activity is NOT observed but the caregiver reports that the child can complete that activity normally, how is it scored?

What is the subsample of children who will have the concurrent measure of development? How is that number established?

Sample Size

The sample size is justified based on a power analysis. The intent is to observe DIF of 1 logit or greater 90% of the time. This degree of DIF is very large – or at least it appears to be. The figure is quoted with no context. What is the intended/likely variance of the latent trait? ETS provides some commentary of the magnitude of DIF (Zwick, 2012) in their use of MH statistics, and in general an ES of 0.5 is considered moderate DIF – it is not clear what their would mean in scale units/logits on the GSED scale. In addition, it should be noted that power analysis reported assumes perfect reliability, which is unlikely. A correction factor unreliability may also be warranted (Adams, 2005). The authors should consider whether the study is adequately powered – it likely is – consider the field trial samples of large-scale assessments (where often the standard of 200 responses per item within each group of interest is considered adequate), however this section is not currently convincing.

Statistical analysis

The authors refer to methods and statistics they will use (e.g., Rasch model, infit/outfit etc) but do not cite relevant literature – consider adding citations that would allow the reader to understand the derivation and application of these measures. For example (Wright & Masters, 1982; Wright & Stone, 1979). Further, the authors describe that they will use R and related libraries (Shiny). These should also be cited – e.g., (R Core Team, 2020).

The authors describe using the "expected a posteriori (EAP) method". See the next paragraph – it is not clear if all readers will accept this is congruent with the fitting of a Rasch model. EAPs are drawn from a model where individual raw scores are factored out and instead quadrature methods are used to integrate over a posterior given by some prior distribution (or subgroup distribution depending on the latent regression model) with some parametrisation (often normal with some sample based mean and variance) and the likelihood function from the item response model. Such combined item-response and population models are extensions of the Rasch model (which is estimated using only the item responses). In addition, see the next section, EAPs are known to be biased, it is not clear why the authors do not use PVs which are standard in LSAs (Marsman et al., 2016; Monseur & Adams, 2009; Wu, 2005). Regardless, the authors should specify the population model they intend to fit to the data, whether it be a single population or a more complex regression model – with much contextual data being collected, there is significant opportunity to impose a large population model.

	The authors describe the item response model they will fit to the data as a Rasch model. Although it is clear what they intend to do, and that the Rasch model yields useful metrics for a validation study (item fit statistics, DIF analysis, standardised residuals etc) it is worth considering being more specific about the model to be fit to the data. For example, refer to the model as a Rasch-like model, or a more general IRM like the one-parameter logistic model. Alternatively, invest more time into justifying why the Rasch model is justified and the limitations of it. Whilst many consider the Rasch model a special case of an item response model (IRM) (for example with appropriate constraints, and estimated using joint maximum likelihood, the Multidimensional Random Coefficients Multinomial Logit Model (Adams et al., 1997) is equivalent to the Rasch Dichotomous Model, others consider the Rasch model as fundamentally different to IRMs with different perspectives on measurements as a science (Andrich, 2004). Regardless, the GSED is characterised by the authors as a population measure, and in this context, the authors are interested in recovering unbiased estimates of population parameters and will therefore want to draw plausible values from case posterior distributions (that is, fit an IRM that parameterises the population distribution) rather than use individual point ability estimates (that is, include an estimation step that calculates individual abilities of each participants ability and report W/MLEs which lead to biased population parameter estimates (Wu, 2005).
--	---

REVIEWER	Jenny Woodman UCL, Social Research Institute
REVIEW RETURNED	16-May-2022

GENERAL COMMENTS	This is a ambitious and important study which has clearly been well-thought out, is articulated well and should be published in this journal so that others can a) see it's happening and b) authors and others can refer back to methods when results are published. I don't feel qualified to comment on the statistical approach here but the aims, objectives and methods are clear. My main comment is about the intended purpose of the GSED. The authors state that the GSED might be used for population monitoring of child development and also 'programme evaluation'. What about use of GSED as an assessment tool to trigger service provision or additional support for an individual child? We know that even if child development measures are not supposed to be used like this, they are (ASQ for example in England). Will the study contextualise the validation of GSED by reporting on potential for implementation in each country and how far this will be / should be used an an individual assessment tool. It would also be helpful to know whether this tool will be free or licensed. Additionally, a very minor point: the 7 countries should be listed in the abstract - to help with those searching for evidence. Thanks you for the opportunity to read this manuscript
--

VERSION 1 – AUTHOR RESPONSE

Reviewer: 1 Dr. Anita D'Aprano, The University of Melbourne

Overall comment. This manuscript describes a study protocol validating the global scales for early development (GSED) in seven countries. The creation of a measure that monitors children's early development is a priority and this is much needed research. The study proposes to recruit 1248 children per country and fit an item response model (specifically Rasch model) to the data to yield an interval measure (D-Score). The paper is very well written and clear.

Thank you we appreciate your positive feedback.

Introduction. Clearly establishes the need for this measure and research.

Thank you we appreciate your positive feedback.

Aims. Clear aim and detailed objectives that explain how the aim will be achieved.

Thank you we appreciate your positive feedback.

Methods. Patient and Public involvement: Co-designed with participants. However, the participants are not described. Is this the caregiver or the services using the measure? This needs elaboration. The participants to the study are caregivers and children. For the involvement in the study design caregivers were engaged as described in the dedicated section which has been modified to clarify the above.

Methods. Study sample: Inclusion criteria does not provide a justification for age up to 41 months.

How was this age determined?

The GSED measures will be validated for children 0 to <36 months. However, to ensure that parameters are estimated with adequate precision for children at the top of our age range and that no ceiling effect is seen, the measures will be tested on children up to 41 months. The study sample section has been modified to reflect such explanation on lines 239-241: "The small sample of children from 36-41 months aims to ensure that parameters are estimated with adequate precision for children at the top of our age range (36 months)."

Methods. Recruitment and consent: It is not clear if families could be approached from being identified through birth registries. Are their details shared with research team? Is this a breach of privacy? How are families with children attending local child health/care centres approached? In a general 'advertisement' inviting participation? This needs further detail to ensure the process is ethical.

Thank you for your comment and we understand the clarification need to avoid concerns. All processes planned for the study are in compliance with local and WHO ERC approval (as indicated in dedicated section on the manuscript). We have added the following text to the manuscript on pages 12-13, lines 250-257 to ensure the processes are clear: "Sites using registries will rely on hospital or health center staff (unaffiliated with GSED) to contact families and obtain consent for sharing their information with the GSED team. A sample listing of the pre-consented families will be provided to the GSDE team for recruitment. Sites recruiting families from local child health/care centers will rely on advertisements or flyers with information about the project, participation requirements, GSED team contact information for questions, and a scan code or website link for interested families to provide basic eligibility information and consent to be contacted for enrollment."

Methods. Sampling frame and schemes: indicate here that the supplementary file has this information.

Thank you for the comment. However, this is already included in the text on line 271: "See sampling Table S1 in Supplementary file S1 for sampling frame"

Data Collection. The GSED LF is described as capturing similar domains to the SF, but observed by the assessor. If the activity is NOT observed but the caregiver reports that the child can complete that activity normally, how is it scored?

We are most grateful to the reviewers for this comment and query. As per similar direct assessment tools, LF items have been designed to be observed incidentally or elicited or both – depending on the item. It is true that if the item is not observed but the caregiver reports the child can complete the activity, it is still scored as a ‘no’. We have added a sentence to the text on line 310-312 on page 15 to clarify this. “LF items must either be observed incidentally or by eliciting the behaviour or both, depending on the item.” This is common practice in this type of assessments and it is taken into account in the scoring precision and therefore interpretation (which is at the group level).

Data Collection. What is the subsample of children who will have the concurrent measure of development? How is that number established?

Thank you for your question. We have added the following text on line 281-284 page 14 in the manuscript for clarification. “For concurrent validity, to assess the GSED against the Bayley-III, a sample size of $N = 150$ per country produces a two-sided 95% confidence interval 0.15-0.44, when the estimate of Pearson’s product-moment correlation is 0.3, with an equal spread of participants tested across age and sex.”

Sample Size. The sample size is justified based on a power analysis. The intent is to observe DIF of 1 logit or greater 90% of the time. This degree of DIF is very large – or at least it appears to be. The figure is quoted with no context. What is the intended/likely variance of the latent trait? ETS provides some commentary of the magnitude of DIF (Zwick, 2012) in their use of MH statistics, and in general and ES of 0.5 is considered moderate DIF – it is not clear what their would mean in scale units/logits on the GSED scale. In addition, it should be noted that power analysis reported assumes perfect reliability, which is unlikely. A correction factor unreliability may also be warranted (Adams, 2005). The authors should consider whether the study is adequately powered – it likely is – consider the field trial samples of large-scale assessments (where often the standard of 200 responses per item within each group of interest is considered adequate), however this section is not currently convincing. See point by point response below.

The sample size is justified based on a power analysis. The intent is to observe DIF of 1 logit or greater 90% of the time. This degree of DIF is very large – or at least it appears to be. The figure is quoted with no context. What is the intended/likely variance of the latent trait?

Thank you. this is a very good question. In this context, it doesn’t make sense to talk about the variance of the trait as, given the fact that age is so closely correlated with ability, the variance of the trait would be highly dependent on the age profile of the sample. We have added the following text, to explain text on line 391-396 page 21: “Given the rapidity of development of children at this age, the latent trait is longer than tends to be found in educational tests which focus on a narrower ability range. The easiest item in our tool “Does your child smile?” has a difficulty of -13.2 logits (1.1 on the D-score scale) and the most difficult item has a difficulty of 8.4 logits (88.86 on the D-Score scale), a 21.6 logit span. Thus, a one logit difference is not particularly large, given the length of the latent trait.” ETS provides some commentary of the magnitude of DIF (Zwick, 2012) in their use of MH statistics, and in general and ES of 0.5 is considered moderate DIF – it is not clear what their would mean in scale units/logits on the GSED scale.

Thank you, we are aware of the ETS guidelines, but for reasons stated above, i.e., long length of the trait the guidelines do not apply.

In addition, it should be noted that power analysis reported assumes perfect reliability, which is unlikely. A correction factor unreliability may also be warranted (Adams, 2005). The authors should consider whether the study is adequately powered – it likely is – consider the field trial samples of

large-scale assessments (where often the standard of 200 responses per item within each group of interest is considered adequate), however this section is not currently convincing.

Thank you for your comment. The study is primarily powered on the probability of finding DIF between two items that have a 1 logit difference, on the latent scale. This is not related to the estimates of ability and their inherent reliability, so unfortunately, we are not clear we understand the point raised.

Statistical analysis. The authors refer to methods and statistics they will use (e.g., Rasch model, infit/outfit etc) but do not cite relevant literature – consider adding citations that would allow the reader to understand the derivation and application of these measures. For example (Wright & Masters, 1982; Wright & Stone, 1979). Further, the authors describe that they will use R and related libraries (Shiny). These should also be cited – e.g., (R Core Team, 2020).

Thank you for your feedback. Additional relevant citations (as per below) have been added to the manuscript:

For Rasch/infit/outfit : Wright BD, Masters GN. Rating Scale Analysis: Rasch Measurement. Chicago: MESA Press; 1982

For R : R Core Team (2016). R: A language and environment for statistical computing. R Foundation for Statistical Computing, Vienna, Austria. URL <https://www.R-project.org/>.

For Shiny: Chang W, Cheng J, Allaire J, Sievert C, Schloerke B, Xie Y, Allen J, McPherson J, Dipert A, Borges B (2022). shiny: Web Application Framework for R. R package version 1.7.2, <https://CRAN.R-project.org/package=shiny>.

Statistical analysis. The authors describe using the “expected a posteriori (EAP) method”. See the next paragraph – it is not clear if all readers will accept this is congruent with the fitting of a Rasch model. EAPs are drawn from a model where individual raw scores are factored out and instead quadrature methods are used to integrate over a posterior given by some prior distribution (or subgroup distribution depending of the latent regression model) with some parametrisation (often normal with some sample based mean and variance) and the likelihood function form the item response model. Such combined item-response and population models are extensions of the Rasch model (which is estimated using only the item responses). In addition, see the next section, EAPs are known to be biased, it is not clear why the authors do not use PVs which are standard in LSAs (Marsman et al., 2016; Monseur & Adams, 2009; Wu, 2005). Regardless, the authors should specify the population model they intend to fit to the data, whether it be a single population or a more complex regression model – with much contextual data being collected, there is significant opportunity to impose a large population model.

Thank you for your comment. We opted for the EAP estimator for two reasons. First, our instruments need to measure a huge variation in proficiency across ages. The EAP method provides a simple and natural way to specify an age-dependent prior distribution for each child. Second, our scales are often short, so there can be many null and perfect scores, which do not translate into a defined estimate under the Rasch model. Thanks to the prior distribution, the EAP method provides a well-interpretable proficiency estimate in these cases. Since the EAP method uses the thresholds estimated under the Rasch model, we would say that its estimates are congruent with the Rasch model.

We did not pursue the Plausible Value (PV) approach in our primary objective: to quantify the child’s level of development and indicate its standard error of measurement. Prospective practitioners would be quite surprised to find that the measurement consists of multiple values from a plausible distribution. Nevertheless, in cases where plausible values are desired, one could easily sample them from the child’s posterior ability distribution as produced by the EAP method. See also response below to the next comment.

We have added the text as follows on line 410-412 page 20: "The method uses a prior normal distribution with a mean set equal to the average proficiency at the child's age and a standard deviation of 5."

Statistical analysis. The authors describe the item response model they will fit to the data as a Rasch model. Although it is clear what they intend to do, and that the Rasch model yields useful metrics for a validation study (item fit statistics, DIF analysis, standardised residuals etc) it is worth considering being more specific about the model to be fit to the data. For example, refer to the model as a Rasch-like model, or a more general IRM like the one-parameter logistic model. Alternatively, invest more time into justifying why the Rasch model is justified and the limitations of it. Whilst many consider the Rasch model a special case of an item response model (IRM) (for example with appropriate constraints, and estimated using joint maximum likelihood, the Multidimensional Random Coefficients Multinomial Logit Model (Adams et al., 1997) is equivalent to the Rasch Dichotomous Model, others consider the Rasch model as fundamentally different to IRMs with different perspectives on measurements as a science (Andrich, 2004). Regardless, the GSED is characterised by the authors as a population measure, and in this context, the authors are interested in recovering unbiased estimates of population parameters and will therefore want to draw plausible values from case posterior distributions (that is, fit an IRM that parameterises the population distribution) rather than use individual point ability estimates (that is, include an estimation step that calculates individual abilities of each participants ability and report W/MLEs which lead to biased population parameter estimates (Wu, 2005).

Thank you for your comments. Our main objective is to develop and validate two measurement instruments that quantify the child's level of development: the GSE short and long form (line 118: "The primary aim of this protocol is to validate the GSED SF and LF."). For this goal, we would like to have a single measured value and an indication of its standard error of measurement. One of the secondary objectives is to create preliminary reference scores for the GSED SF and LF (line 119).

Although not explicitly stated, the reviewer alludes to the problem that using a single, best value for the D-score will underestimate the variability in the references. We agree that this is a problem. We also agree that adding a PV step, as suggested by the reviewer, would deliver a better estimate of the population reference as it accounts for measurement error. However, we did not consider the use of PV for three reasons. First, it is unusual in the field (for example we know of no anthropometric references that implement PV's). Second, it is incorrect to compare a set of EAP estimates with PV-derived references since the EAP estimates would not sufficiently cover the tails. Third, doing the correct process (implementation of full PV approach) for individual EAP estimates is highly impractical. We, therefore, thank the reviewer for the suggestions, but find that the proposed approach is still the most more practical and conforms to standard practice. Since the "best value" (EAP) does not reflect the uncertainty of the measurement (which is always there), the variability of the population reference is less than a hypothetically correct method that includes measurement error. The usual case is to ignore the measurement error, and we also conform to that practice.

However, in order to address the concerned raised, we have added the text as follows on line 414-419 page 21 "Note that this application of EAP estimates underestimates the true variability in the population because EAP estimates – as any measurement – are always imprecise. In daily practice, analysts will compare other EAP estimates to the reference. To support this type of application, we create the references from the EAP estimates and accept a (perhaps slight) underestimate of the true variability in child development in the population."

We have also revised the text as follows on line 533 page 27 "This univariate model makes strict assumptions designed for global population estimates and may exclude items that do not show strong age gradients or items that measure development in a culturally-specific ways"

Reviewer: 2 Miss Jenny Woodman, UCL

Comments to the Author: This is a ambitious and important study which has clearly been well-thought out, is articulated well and should be published in this journal so that others can a) see it's happening and b) authors and others can refer back to methods when results are published. I don't feel qualified to comment on the statistical approach here but the aims, objectives and methods are clear. Thank you we appreciate your positive feedback.

My main comment is about the intended purpose of the GSED. The authors state that the GSED might be used for population monitoring of child development and also 'programme evaluation'. What about use of GSED as an assessment tool to trigger service provision or additional support for an individual child? We know that even if child development measures are not supposed to be used like this, they are (ASQ for example in England). Will the study contextualise the validation of GSED by reporting on potential for implementation in each country and how far this will be / should be used as an individual assessment tool. It would also be helpful to know whether this tool will be free or licensed.

We are grateful for this comment by the reviewers and agree that it is important to consider how the tool might be used as an assessment tool as they often are, even if this is not their intended use. GSED is a free, open access tool available for global use. However, it was designed and being validated to fill a gap for population and programmatic assessments using a very rigorous and culturally informed process. It has not been developed or evaluated for use with individual children. because we cannot assure the sensitivity and specificity of the GSED as an individual measure, we do not recommend it as a screening nor a diagnostic measure for interpretation at individual level and we are very clear in all GSED related materials and administration manuals. We recognize the desire for a global screening measure to apply to individual children to ensure that children receive needed interventions. As this question is so important, we have already gained further funding to move in the direction of understanding how the GSED package could be modified to be validated for such purpose through a rigorous processl.

We have added a sentence to the text on line 545-548 on page 27 to clarify this. "Lastly, as the GSED SF and LF scores are meant to be interpreted and used for population-level measurement, we plan to expand the work towards understanding of how the GSED package could be modified and validated to be able to identify individual children at risk of developmental delays and disorders".

Additionally, a very minor point: the 7 countries should be listed in the abstract - to help with those searching for evidence.

Thank you for this comment. We have added the names in the abstract.

Thanks you for the opportunity to read this manuscript

Thank you for your comments, we very much appreciate you taking the time to read our work.